# Salt-Related Knowledge, Attitudes and Behaviors (KABs) among Victorian Adults Following 22-Months of a Consumer Awareness Campaign

**DOI:** 10.3390/nu12051216

**Published:** 2020-04-26

**Authors:** Carley A. Grimes, Durreajam Khokhar, Kristy A. Bolton, Kathy Trieu, Jane Potter, Chelsea Davidson, Elizabeth K. Dunford, Stephen Jan, Mark Woodward, Bruce Bolam, Bruce Neal, Caryl Nowson, Jacqui Webster

**Affiliations:** 1Institute for Physical Activity and Nutrition, Deakin University, Geelong 3220, Australia; carley.grimes@deakin.edu.au (C.A.G.); a.khokhar@deakin.edu.au (D.K.); caryl.nowson@deakin.edu.au (C.N.); 2School of Exercise and Nutrition Sciences, Deakin University, Geelong 3220, Australia; 3The George Institute for Global Health, University of New South Wales, Sydney 2050, Australia; ktrieu@georgeinstitute.org.au (K.T.); edunford@georgeinstitute.org.au (E.K.D.); sjan@georgeinstitute.org (S.J.); markw@georgeinstitute.org.au (M.W.); bneal@georgeinstitute.org.au (B.N.); jwebster@georgeinstitute.org.au (J.W.); 4Victorian Health Promotion Foundation (VicHealth), Melbourne VIC 3053, Australia; jpotter@vichealth.vic.gov.au; 5Heart Foundation, Melbourne, Victoria 3000, Australia; Chelsea.Davidson@heartfoundation.org.au; 6Department of Nutrition, The University of North Carolina, Chapel Hill, NC 27599, USA; 7The George Institute for Global Health, University of Oxford, Oxford OX1 2BQ, UK; 8Department of Health and Human Services, Melbourne, Victoria 3000, Australia; bruce.bolam@dhhs.vic.gov.au

**Keywords:** salt, sodium, knowledge, attitude, behavior, parents, Australia

## Abstract

The Australian population consumes more salt than recommended and this increases the risk of raised blood pressure and cardiovascular disease. In 2015, a state-wide initiative was launched in the Australian state of Victoria to reduce population salt intake. This study examines whether salt-related knowledge, attitudes and behaviors (KABs) of Victorian adults changed following the first 22 months of a consumer awareness campaign targeting parents. Repeated cross-sectional surveys of adults (18–65 years) recruited from research panels. Analyses were weighted to reflect the Victorian population. In both surveys mean age of participants (1584 in 2015 and 2141 in 2018) was 41 years, and 51% were female. This includes 554 parents/caregivers in 2015 and 799 in 2018. Most indicators of KAB remained unchanged. Among parents/caregivers the percentage who agreed limiting salt in their child’s diet was important increased by 8% (*p* = 0.001), and there was a 10% reduction in the percentage who reported placing a saltshaker on the table and a 9% reduction in those who reported their child added salt at the table (both *p* < 0.001). Some small adverse effects on other indicators were also observed. During the first 22 months of a salt reduction consumer awareness campaign, there were limited changes in KAB overall, however the target audience reported positive changes regarding their children, which aligned with the campaign messages.

## 1. Introduction

Reducing the amount of salt in people’s diets is considered a ‘best-buy’ public health intervention for the prevention of non-communicable disease [1], because excess salt intake is associated with raised blood pressure and cardiovascular disease [2]. Globally, it is estimated that adults consume, on average, 10.1 g of salt per day [3]. The situation is similar in Australia where mean daily intakes, after adjusting for non-urinary salt excretion, are estimated as 9.6 g/day (10.1 g/day in males and 7.3 g/day in females) [4]. However, dietary guidelines specify that adults should consume no more than 5 g/day of salt [5,6]. Intake of salt is also high among children, with approximately three quarters of Australian children consuming more than is recommended [7,8]. 

In Westernized food systems, most salt in the diet (~75%–85%) comes from that added to processed and restaurant foods during manufacture. The remainder (~15%–25%) comes from salt added by the consumer during cooking or at the table [9,10]. To reduce salt intake, a comprehensive range of strategies is recommended. These include reformulation of food products to contain less salt, front-of-pack labelling, provision of lower salt options in public institutions (e.g., schools and workplaces) and consumer education campaigns [1,11]. The World Health Organization (WHO) has set a global target for a 30% reduction in population intake of salt by 2025 [1]. Australia has adopted this target [12], and at the state-level within Victoria there is a strong commitment to reducing population salt intake [13]. In 2015, the Victorian Salt Reduction Partnership (referred to as The Partnership) launched a multi-component state-wide initiative designed to reduce average salt intake in Victorian children and adults by 1 g/day. The Partnership, led by the Victorian Health Promotion Foundation (VicHealth) is comprised of a group of stakeholders from health-related non-governmental organizations, the state government and the academic sector. The Partnership governed implementation of three key streams: (1) raising consumer awareness to improve attitudes and change behaviors, (2) strengthening policy initiatives, supporting food industry innovation and (3) ongoing rigorous research, monitoring and evaluation [14]. 

VicHealth, in conjunction with the Heart Foundation (Victoria), led the development and implementation of the consumer awareness campaign component of this state-wide initiative. The primary target audience for the campaign was Victorian women with children 0–12 years who were primary or joint household grocery buyers. The first phase of the campaign titled “Don’t Trust Your Taste Buds” ran for 6 weeks during June and July 2016 [15]. This phase of the campaign aimed to raise awareness surrounding high salt consumption within the Victorian population, the link between excess salt and adverse health outcomes, particularly for children, and the hidden sources of salt in the food supply. Phase two of the campaign titled “Unpack the Salt” ran for 21 months from August 2017 to April 2019. During this time there were five waves of planned activities (Appendix A). Throughout this phase, key messages from phase one were reiterated along with top tips to reduce salt in the diet (e.g., reading sodium information on food labels to select foods with less salt, choosing the right packaged food, limiting salt use at the table and during cooking) along with the promotion of fresh foods and low salt recipes [16]. In the final stages of the campaign, this was further supported with an additional focus on behavior change strategies to reduce salt intake. This included signing up to the “Unpack Your Lunch Campaign” which was a 10-day salt challenge targeting actions to lower salt intake [16]. Static and digital advertisements, campaign website and social media advertisements were the primary paid channels of communications. Unpaid exposure from the campaign was generated via media coverage [14]. To evaluate the effectiveness of the consumer awareness campaign, repeated surveys assessing knowledge, attitudes and behaviors (KAB) related to salt intake were conducted on a sample of the Victorian population [14]. The aim of this study was to determine if salt-related KABs of Victorian adults changed following the first 22 months of a salt reduction consumer awareness campaign. 

## 2. Materials and Methods 

### 2.1. Study Design and Participants

This was a repeated cross-sectional online survey of Victorian adults aged 18–65 years. The survey was conducted at two time points: at baseline in November 2015 (referred to as time point 1; T1) and in March-April 2018 (referred to as T2). T2 aligned with the approximate mid-point of the planned consumer awareness campaign (22 months of implementation). Participants were recruited via a research panel company (Lightspeed Research, Melbourne, Australia). The Lightspeed Research database is a database of individuals who have voluntarily registered themselves with the company, and are periodically contacted to take part in a variety of online surveys in return for reward points, which they can redeem for monetary payments. Due to logistical difficulties with the research company’s sampling framework, it was not possible to source two independent samples at each time point, and it was anticipated that approximately 10% of the final sample would contain repeated participants. Quotas were set for recruitment based on age and sex groups that reflected the Victorian population [17]. Participants aged over 65 years were excluded, on the basis that salt-related public awareness initiatives would primarily target those aged under 65 years. The survey was administered using Qualtrics survey software. All participants provided informed consent and ethics approval was obtained by the Deakin University Human Ethics Advisory Group (Project No: HEAG-H 83_2015 and Project No: HEAG-H 71_2016). 

### 2.2. Survey Instrument

A 37-item questionnaire was developed to assess demographic characteristics and KAB related to dietary salt intake. Demographic characteristics assessed included age, sex, country of birth, language spoken at home, residential postcode, educational attainment and responsibility for household grocery shopping. Participants self-reported on weight, height, diagnosis of a chronic condition, use of antihypertensive medication and if they had previously received advice from a health professional to limit salt intake. Based on educational attainment participants were grouped as low, mid and high socioeconomic background (SES) (low: includes those with some or now level of high school education, mid: includes those with a technical/trade certificate or diploma, high: includes those with a university/tertiary qualification). Body mass index was calculated and participants were grouped into weight categories [18]. The KAB questions were modelled on those used in previous salt related surveys [13,19,20,21,22,23,24,25,26,27,28,29,30,31]. All questions and response options used at each time point are shown in Appendix A, along with reference to where the questions were sourced. The questionnaire was pilot tested with 20 adults of varying demographic background (age, sex, SES). From this, minor revisions were made to improve readability and reduce survey completion time. 

Salt-related knowledge was assessed with six questions, one of which contained four separate items. These included knowledge of the relationship between salt and sodium, the main source of salt in the diet, dietary recommendations for salt intake, how much salt Australians consume and health outcomes associated with high salt intake. Categorical response options varied across questions (see Appendix A). Attitudes related to salt intake were assessed with four questions, one of which contained six separate items. These related to how participants viewed their own intake of salt compared to recommendations, their level of concern for a range of food related issues (e.g., healthy eating, the amount of sugar, salt and fat in food) and who they believed was responsible for reducing population salt intake (e.g., food manufacturers, the government, yourself). The final question assessed agreement with a series of six attitude statements e.g., ‘my health would improve if I reduced the amount of salt in my diet’, ‘I believe salt needs to be added to food to make it tasty’. A range of categorical and Likert scale response options were used. Salt related behaviors were assessed with five questions, one of which contained seven separate items. Three questions related to discretionary salt use e.g., salt use during cooking, at the table and if a saltshaker was placed on the table during mealtimes. Responses ranged from ‘always’ to ‘never’. Participants were asked if they were trying to cut down on the amount of salt in their diet (responses: ‘yes’, ‘no’, ‘don’t know’) and what behaviors they had engaged in within the previous month to reduce dietary salt (e.g., ‘used spices/herbs instead of salt when cooking’ (responses: ‘never do this’ to ‘always do this’ and ‘does not apply to me’). One question assessed participants knowledge of the state-wide salt reduction campaign ‘Are you aware of VicHealth’s initiative to reduce salt intake within the Victorian population?’ 

Participants that identified as being a parent or caregiver for at least one child under the age of 18 years were asked an additional nine questions, as part of a pre-specified plan to analyze this subgroup separately. One question assessed the age of their children, and the remaining eight questions assessed KAB related to children’s salt intake. Specifically, two questions assessed knowledge of how much salt Australian children eat, and the long-term health effects of eating too much salt during childhood. Two questions assessed attitudes related to the importance of limiting salt in their child’s diet and if more action should be taken to reduce salt in foods targeted at children (responses: ‘strongly disagree’ to ‘strongly agree’). Three questions assessed discretionary salt use behaviors of their child/ren (e.g., child’s use of salt at the table, use of salt when cooking meals prepared for children, and placement of a saltshaker on the table during mealtimes). Responses ranged from ‘always’ to ‘never’. 

### 2.3. Data Analysis

All data were analyzed using Stata/SE 15 (StataCorp LP, College Station, TX, USA). Descriptive statistics, mean or percentage and 95% confidence intervals (CI) were used to describe participant characteristics and survey responses. Post-stratification weights were created to weight the data to reflect the Victorian population of adults aged 18–65 years for sex and age-group [32] using the probability weight (pweight) in Stata. All presented results relate to weighted estimates. Although the final analytical sample contained *n* = 195 (5.3%) repeated participants at each time point, data were analyzed as independent samples. The error associated with this approach is reduced variability, which would marginally compromise power and increase the likelihood of Type II error.

For analyses, question responses were generally dichotomized. For example, knowledge responses were grouped as either correct vs. incorrect, level of agreement with attitude statements responses were grouped as ‘strongly agree/agree’ vs. ‘strongly disagree, disagree, neither agree nor disagree’, discretionary salt use behaviors were grouped as ‘always/often/sometimes’ and ‘rarely/never’, other salt reduction behaviors were grouped as ‘never do this’/’rarely do this’ vs. ‘always do this’/‘often do this’/’sometimes do this’. For one question (‘how do you think your daily salt intake compares to the amount recommended by health professionals’), the dichotomizing of response options was nonsensical, so response options remained separated. To assess differences in categorical survey responses assessed at T1 compared to T2 Pearson’s chi-squared tests were used. 

Multivariate logistic regression models were used to assess whether differences remained significant after adjustment for covariates. A *p*-value of <0.05 was considered statistically significant. The selection of covariates was based on participant demographic characteristics that differed between T1 and T2 (*p* < 0.20), and these criteria excluded age or sex, which were already weighted for in the analysis. Language spoken at home (*p* = 0.003) was not included, as this variable was associated with country of birth, and country of birth was selected as the marker of ethnicity. Other specific conditions (e.g., heart disease, stroke, high blood pressure) were not included, as instead, the collective diagnosis with any chronic condition was used. The result was that in the total sample, models were adjusted for country of birth and diagnosis with a chronic condition. In analyses restricted to the sub-sample of parents/caregivers, models were adjusted for those characteristics that differed (*p* < 0.20) within this select group. This comprised country of birth, SES, diagnosis with a chronic condition and responsibility for household grocery shopping. 

## 3. Results

Across both time points 3943 participants (1655 in 2015 and 2288 in 2018) agreed to complete the online survey (response rate 14% in 2015 and 12% in 2018). Participants who reported a residential postcode outside of the state of Victoria, Australia (*n* = 88) or who did not complete the survey to the end (*n* = 130) were excluded from analysis. The final analytical sample was 3725 (1584 in 2015 and 2141 in 2018). This included a sub-sample of 1357 parents/caregivers of children under 18 years of age. Among these, a further four were excluded, as they did not answer all of the additional KAB questions pertaining to their child/ren, leaving a final analytical sample of 1353 (554 in 2015 and 799 in 2018) parents/caregivers (Figure 1). 

### 3.1. Demographic Characteristics of Participants

At both time points the sample was comprised of 51% females with an average age of 41 years. Overall, there was a reasonable spread of participants by socioeconomic background, and just over two thirds of the sample were identified as being responsible for the household grocery shopping (Table 1). Less than a fifth of participants were aware of VicHealth’s initiative to reduce salt intake within the Victorian population at both time points (15% at T1 and 18% at T2, *p* = 0.007).

### 3.2. Change in Knowledge Related to Salt Intake among Victorian Adults between 2015 (T1) and 2018 (T2)

Overall salt-related knowledge (9/10 questions) remained unchanged between T1 and T2 (Table 2). At both time points, the majority of participants were aware that Australians eat too much salt (~80%) and that excess salt could damage health (~90%), whereas considerably fewer (~one third) could correctly identify the relationship between salt and sodium or the daily salt intake guideline. There was a small improvement in the percentage of adults (+3.7%, *p* = 0.02) who correctly identified processed foods as the main source of salt in the Australian diet (Table 2), and this difference remained in the adjusted analyses. 

### 3.3. Change in Attitudes Related to Salt Intake among Victorian Adults between 2015 (T1) and 2018 (T2)

Overall salt-related attitudes (7/10 questions) remained unchanged between T1 and T2 (Table 2, Appendix A). Across time points there were some small positive improvements in specific attitudes related to salt intake. Firstly, compared to T1, at T2 more participants (+4.2%, *p* = 0.01) agreed with the statement that their health would improve if they reduced the amount of salt in their diet and fewer participants (−5.7%, *p* < 0.001) agreed that it was difficult to understand sodium information displayed on food products (Table 2), both remained significant in adjusted analyses. Conversely, there was no shift in participants’ attitudes related to use of gourmet salts over regular table salt, the availability of lower salt options when eating out or the need for laws limiting the amount of salt added to manufactured foods (Table 2). With regards to participants’ views on how their own daily salt intake compared to that recommended by health professionals, significantly more (+5.5%, *p* = 0.001) believed they consumed salt in excess of dietary recommendations at T2 compared to T1 (Table 2). There was no shift in participants’ level of concern for a range of different food-related issues (e.g., the amount of sugar or fat in food) (Appendix A). This included concern for the amount of salt in the diet, which remained unchanged with ~85% of participants reporting that they were either somewhat, very or extremely concerned with this at both T1 and T2. Overall, there was no shift in participants’ attitudes towards groups in society that they viewed as being responsible for reducing the amount of salt Australians eat (Appendix A). 

### 3.4. Change in Behaviors Related to Salt Intake Among Victorian Adults between 2015 (T1) and 2018 (T2)

Overall, six out of the eleven assessed salt related behaviors changed between T1 and T2, albeit in the opposite direction than what was intended from implementation of the salt reduction program (Table 2). Specifically, compared to T1, at T2 more participants reported adding salt during cooking (+3.8%, *p* = 0.02); however, this finding was no longer significant in adjusted analyses (*p* = 0.06). In addition, fewer participants reported engaging in a number of favorable behaviors that could lead to the consumption of less salt (e.g., avoiding eating packaged foods, avoiding eating from fast food restaurants or Asian style restaurants, purchasing foods labelled salt-reduced or requesting for a meal to be prepared without salt when eating out), all of which remained in adjusted analyses. There was no change in the percentage (~40%) of participants who reported trying to cut down on the amount of salt in their diet between time points. Similarly, there was no change in the percentage of participants who reported use of table salt (~50%) or placing a saltshaker on the table (~50%). 

### 3.5. Change in Knowledge, Attitudues and Behaviors Related to Salt Intake Among Victorian Parents/Caregivers between 2015 (T1) and 2018 (T2)

Demographic characteristics of the sub-sample of parents/caregivers at each time point are shown in Appendix A. Among the sample of parents/caregivers there was no shift in their knowledge that related to salt intake among children (Table 3). Overall, it appeared that the majority of parents/caregivers were already aware that Australian children eat too much salt, and that excess salt intake during childhood was linked to adverse health outcomes. In relation to attitudes, there was a significant increase in the percentage of parents/caregivers, who believed that limiting the amount of salt their child/ren eats was important to them (+8.3%, *p* = 0.001), and this difference remained significant in adjusted analyses. In addition, there was improvement in all discretionary salt use behaviors that related to their child/ren**.** For example, ~10% fewer parents/caregivers reported that they placed a saltshaker on their table during mealtimes, and that their child/ren added salt to their food at the table (Table 3). 

## 4. Discussion

Following the first 22 months of a salt reduction consumer awareness campaign in Victoria, Australia, there was little change in self-reported knowledge, attitudes and behaviors related to salt intake. Within the wider adult population sample, although a few limited markers of knowledge and attitudes improved, this did not translate into favorable changes in behavior to reduce salt in the diet, and instead, a number of salt-related behaviors worsened following the intervention period. Favorable changes were observed among parents/caregivers of children aged 18 years and under, with a greater percentage of parents/caregivers believing that limiting salt in their child’s diet was important, and lower proportions reporting discretionary salt use behaviors pertaining to their children. Women with children aged 0–12 years were a specific target audience for the campaign.

The consumer awareness campaign implemented by VicHealth and the Heart Foundation was designed to raise general awareness about excess salt in the diets of adults and children, as well as to increase salt-related knowledge. Specific areas of knowledge that were targeted during the campaign related to daily recommendations for salt intake, the link between excess salt and adverse health outcomes and the hidden sources of salt in the diet. Findings from the current evaluation indicate that knowledge has remained largely unchanged. It is worth noting that baseline knowledge across some of these areas was already high, and there was little room for improvement. For example, the majority of participants were already aware that Australians eat too much salt, and that this is detrimental to health. This is a common finding within other population groups assessing salt-related knowledge [35], and suggests the need for better targeting of messages to less well informed segments of the population by future awareness campaigns. In comparison, at baseline, relatively few participants (less than a third) could correctly identify the recommended daily limit for salt of 5 g, a finding consistent within the literature [35], and this remained unchanged at follow-up. Although this was a key message disseminated during the campaign it does not appear to have reached the target population group. The only knowledge item that did show improvement, albeit a small one, was that which related to correctly identifying processed foods as the main source of salt in the Australian diet. This was a key message within the campaign and messages surrounding the hidden sources of salt in commonly consumed processed foods (e.g., cooking sauces, condiments, frozen meals and processed meats) were repeated throughout the campaign. A key campaign tag line was “If it’s packed (e.g., packaged food) chances are it’s packed with salt”. This message was supported with regular media releases, which identified the saltiest foods within selected food categories [36,37], and frequently gained widespread coverage. For example, coverage related to the high salt content of sausages reached an audience of 7,460,925, with 274 media items via all national TV news stations, along with major state-based newspapers and radio programs and 143 #UnpackTheSalt mentions on social media. It will be of interest to assess if further uptake of this knowledge item has occurred by a wider group of the population at the end point evaluation period of the public awareness campaign. 

There was a 6% decrease in the percentage of participants who reported difficulty in understanding sodium information displayed on food products. Education materials on this topic were disseminated throughout the campaign. Other studies frequently report that consumers find sodium information on food labels difficult to understand [35]. It is anticipated that educating consumers to accurately decipher labelled information can aid them in reading the label to make better food choices which contain less salt. 

There was also a small 6% increase in the percentage of adults who believed that their own salt intake exceeded recommendations. This finding may reflect some improvement in self-awareness of their own personal dietary intake, and awareness that the issue of high salt intake applies not only to the wider community but also themselves. This is also supported by the finding that, at T2, more participants believed their health would improve if they reduced the amount of salt in their diet. These shifts in a small section of the population may indicate that some participants understood the direct personal consequences of a salty food supply, and the potential benefits that could be gained by cutting back on salt. This is an important first step on the pathway to achieving behavior change, and it is well recognized that targeting personal attitudes and beliefs is an important component of behavior change programs [38,39]. However, despite this, we did not see any favorable changes in reported salt reduction behaviors. Instead, we observed no change in those who reported trying to limit salt in their diet (~40% at both time points) and no change in discretionary salt use behaviors. At T2, contrary to the expectation that salt-related behaviors would improve, we observed a small, yet significant, reduction in the percentage of participants (~4%) who reported engaging in behaviors that would limit salt in their diet (e.g., purchasing salt-reduced foods, avoiding eating packaged foods or from fast food restaurants). It is difficult to explain this finding, particularly in light of the slight positive shifts noted in some attitudes towards salt intake. It is possible that, although the campaign to date showed some success in raising individual’s awareness of their own salt intake, consumers may still lack the capacity, skills or confidence to implement relevant dietary changes. It is also possible that the mid-point evaluation was too early to detect changes in behavior, particularly as targeted messaging on actions to reduce salt in the diet were the focus of the final two waves of the campaign, occurring after the T2 evaluation period (Appendix A). 

Other countries that have implemented a multifaceted, comprehensive salt reduction strategy which has incorporated a consumer awareness campaign include the UK and South Africa [40,41]. Unlike the findings from the present study, evaluation surveys conducted in each of these countries indicate that, following dissemination of the consumer awareness campaigns, there was a positive shift in a number of salt-related KABs [40,41]. For example, in the UK, following the 2004 implementation of the Food Standards Agency’s large-scale salt reduction consumer awareness campaign, the percentage of adults generally adding salt at the table reduced by 9% between 2003 to 2007 [42]. There were also improvements in other salt-related knowledge and attitudes. For example, the percentage of adults aware of the main source of salt increased from 29% to 50%, and those aware of the daily salt intake recommendation increased from 34% to 43% [43]. This contrasts findings from the present study where no changes in these knowledge markers were observed. 

In South Africa, following a one-year (2014-2015) consumer awareness campaign led by the The Heart and Stroke Foundation of South Africa (HSFSA), which included TV and radio advertisements a number of improvements were observed in salt-related KAB markers. This included a significant improvement in the percentage of participants aware that high salt intake is detrimental to health (76% baseline and 89% follow-up) [40], as well as those who considered lowering salt in the diet to be important (67% at baseline to 75% at follow-up). There was also a significant reduction in the percentage of participants who reported adding salt at the table (21% baseline and 15% follow-up) and during cooking (63% baseline and 40% follow-up) [40]. Whilst this South African study was conducted with the same participants at baseline and follow-up, a convenience sample of only black females was included, thereby limiting the generalizability of their findings to the wider population. 

It is important to note that both campaigns from the UK and South Africa have been considerably more intensive than the current VicHealth and Heart Foundation led campaign. At the time of the current mid-point evaluation the Victorian consumer awareness campaign had only been running for 22 months, whilst the UK’s consumer awareness campaign ran for approximately 5 years (2004–2009). Similar to the Victorian campaign, the UK’s campaign involved press, digital advertising and information published on the UK’s Food Standard agency websites, combined with widespread television and radio advertising [41]. 

In the current study, we did find a positive shift in discretionary salt use behaviors of children, as reported by parents/caregivers. At baseline, about half of parents/caregivers reported that they added salt to foods prepared for their child/ren, and that they placed a saltshaker on the table at mealtimes, and approximately a third reported that their child/ren added salt to their food at the table. These estimates are comparable to those reported within a previous sample of Victorian primary schoolchildren [7]. At T2, these estimates shifted downward, with a 6%–10% reduction in the percentage of parents reporting these behaviors in their child/ren. Although discretionary salt contributes (~15%–25%) far less to daily salt intake compared to processed foods (~75%–85%), it is still considered an important component to target within population salt reduction interventions [41]. Findings from a previous small study conducted in Victorian primary schoolchildren showed that those children who reported adding salt at the table had significantly higher 24-h urinary sodium excretion (equivalent to 1 g/day of salt) compared to those who did not [44]. As we did not measure salt intake within the current study, it is unclear if this downward shift in reported discretionary salt use behaviors would have translated to a reduction in children’s salt intake. Information and tips to stop adding salt to food (e.g., salt flavor alternatives) were provided via the Heart Foundation’s Unpack the Salt website. Our findings suggest the beginning of the uptake of these messages by parents. However, it is unclear as to why these same shifts were not observed among the wider population of adults for their own discretionary salt use behaviors. It is possible that among adults, lifelong habits related to their own discretionary salt use may be more challenging to change, therefore reinforcing the need for reformulation efforts to reduce salt added to processed foods. It is also possible that the desire to protect the health or make positive changes to their child/ren’s behavior is higher than that of their own. To our knowledge, no other studies evaluating the impact of consumer awareness campaigns on salt-related KABs have focused on parents. 

Strengths of this study include the use of a large sample size at both time points using identical methods for recruitment, including age group and sex quota targets to help capture a representative sample of the Victorian population. The questionnaire was developed by experts working in the area of salt reduction, with consultation from VicHealth to appropriately tailor it to the planned salt reduction intervention. The questionnaire was modelled on those used in previous surveys [13,19,20,21,22,23,24,25,26,27,28,29,30,31] and pilot tested for readability and comprehension. To our knowledge, during the evaluation period there were no other national initiatives that may have contributed to any of the observed changes in KABs. The study was limited by a response rate of 14%, and, although comparable to the average response rate achieved in online panels of 10%–15% (personal communication, Lightspeed Research), this may introduce non-response bias. As no information was collected on non-responders in this survey, it was not possible to assess this potential bias. The effect of this bias would be limited though, as we assessed differences in KAB markers pre and post intervention, with similar response rates at each time point. Although KAB surveys are frequently used to evaluate the effectiveness of salt reduction interventions [45], it is acknowledged that such surveys are limited to self-reporting bias (e.g., social desirability bias), and there may be differences between behaviors reported and those actually undertaken. There may also be bias in participation as respondents received points for completing the online survey, which they could redeem for monetary payments. 

## 5. Conclusions

Following the first 22 months of a salt reduction consumer awareness campaign in Victoria, Australia, there were limited changes in self-reported knowledge, attitudes and behaviors among Victorian adults. There were anticipated small improvements in some indicators, unanticipated adverse effects for others and no change for many. The strongest evidence of improvement related to the behaviors of children reported by adults. To what extent the observed changes reflect real effects, chance or small biases remains somewhat uncertain, but the apparently limited overall impact of the intervention raises questions about the likely impact on population salt intake. At present, the mid-point evaluation findings indicate that future salt reduction consumer awareness campaigns may need to be more intensive in terms of their dissemination strategies, as well as being prolonged. Furthermore, given the difficulty in shifting consumers behaviors to limit salt in the diet, food reformulation of lower salt foods remains a central and key component of future salt reduction initiatives. It will be of interest to determine which changes are sustained at the conclusion of the evaluation period, and what are the effects on salt consumption. These additional findings can be used to inform the development of future population based strategies to reduce salt consumption. 

## Figures and Tables

**Figure 1 nutrients-12-01216-f001:**
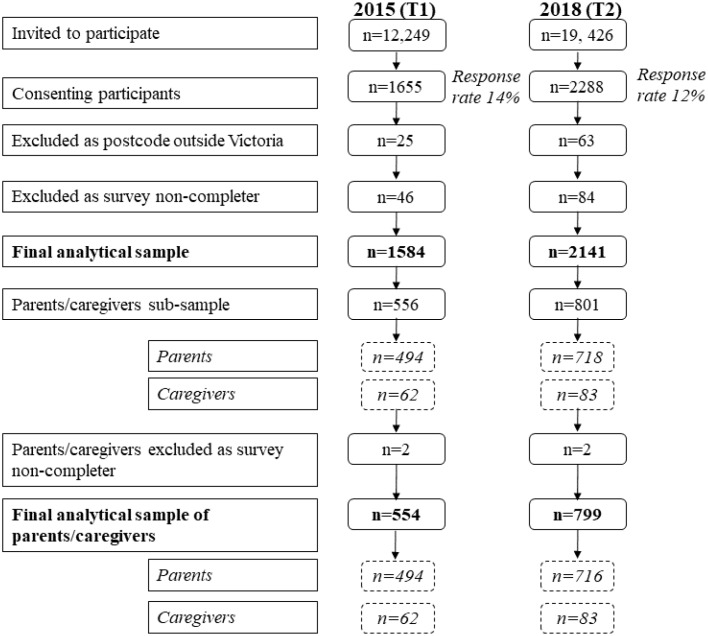
Flow chart of participation in 2015 and 2018.

**Table 1 nutrients-12-01216-t001:** Demographic characteristics of participants, 2015 (*n* = 1584) and 2018 (*n* = 2141).

Characteristic	2015 (T1)			2018 (T2)			*p*-Value ^b^	Victorian Population (%) ^c^
	*n* (Unweighted)	% or Mean (Weighted) ^a^	95% CI	*n* (Unweighted)	% or Mean (Weighted) ^a^	95% CI		
Gender								
Male	843	49.1%	(46.6 51.7)	1063	49.1%	(46.9, 51.3)	0.96	49.1
Female	741	50.9%	(48.3, 53.4)	1078	50.9%	(48.7, 53.1)		50.9
Age (years) (mean)		40.6	(39.9, 41.3)		40.6	(39.9, 41.3)	0.94	
Age group								
18–24 year	156	14.9%	(12.9, 17.2)	254	14.9%	(13.3, 16.7)	0.002	14.9
25–34 year	396	23.6%	(21.6, 25.8)	439	23.5%	(21.6, 25.6)		23.6
35–44 year	399	21.3%	(19.5, 23.3)	526	21.4%	(19.7, 23.1)		21.3
45–54 year	342	20.6%	(18.7, 22.7)	505	20.6%	(18.9, 22.4)		20.7
55–65 year	291	19.5%	(17.5, 21.7)	417	19.6%	(18.0, 21.4)		19.5
Country of Birth								
Australia	1292	81.9%	(79.8, 83.7)	1682	78.0%	(76.1, 79.8)	0.07	
United Kingdom	51	3.2%	(2.4, 4.2)	87	3.9%	(3.2, 4.8)		
New Zealand	18	1.2%	(0.8, 1.9)	40	1.8%	(1.3, 2.5)		
Other	203	12.5%	(10.9, 14.3)	310	15.1%	(13.5, 16.8)		
Don’t know/prefer not to answer	20	1.2%	(0.8, 1.9)	22	1.2%	(0.8, 1.8)		
Do you speak a language other than English at home?								
Yes	257	16.4%	(14.6, 18.4)	390	19.4%	(17.7, 21.2)	0.003	
No, English only	1313	82.8%	(80.7, 84.6)	1713	78.8%	(76.9, 80.6)		
Don’t know/prefer not to answer	14	0.8%	(0.5, 1.5)	38	1.8%	(1.3, 2.5)		
Socio-economic status (SES) based on highest level of education ^e^								
High SES	629	39.2%	(36.7, 41.8)	872	40.3%	(38.2, 42.5)	0.63	28.1% ^d^
Mid SES	475	29.6%	(27.3, 31.9)	604	28.1%	(26.2, 30.1)		27.0%
Low SES	464	31.2%	(28.9, 33.7)	652	31.6%	(29.6, 33.7)		42.9%
BMI (kg/m^2^) (mean) ^f^	1363	26.7	(26.3, 27.0)	1852	27.3	(26.9, 27.9)	0.01	
Weight category ^f^								
Underweight (Body Mass Index (BMI) <18.5 kg/m^2^)	48	4.0%	(3.0, 5.4)	56	3.4%	(2.6, 4.5)	0.27	2.3% ^g^
Healthy weight (BMI ≥18.5–24.9 kg/m^2^)	553	41.5%	(38.8, 44.2)	704	39.3%	(37.0, 41.7)		37.7%
Overweight (BMI ≥25.0–29.9 kg/m^2^)	437	31.2%	(28.8, 33.8)	591	31.0%	(28.9, 33.2)		30.6%
Obese (BMI ≥ 30.0 kg/m^2^)	325	23.3%	(21.1, 25.6)	501	26.2%	(24.2, 28.3)		19.1%
Diagnosed with a chronic condition								
Yes	454	28.2%	(25.9, 30.5)	555	24.9%	(23.1, 26.8)	0.09	
No	1105	70.4%	(68.1, 72.7)	1555	73.4%	(71.4, 75.3)		
Don’t know/can’t recall	25	1.4%	(1.0, 2.1)	31	1.7%	(1.2, 2.4)		
Have you ever been diagnosed with or suffered from one or more of the following conditions? (yes)								
Heart Disease	78	4.7%	(3.7, 5.8)	4.7	3.2%	(2.6, 4.1)	0.03	
Stroke	55	3.2%	(2.4, 4.2)	3.2	1.6%	(1.1, 2.2)	0.001	
Heart attack	46	2.7%	(2.0, 3.6)	2.7	2.4%	(1.8, 0.3)	0.58	
Other (please specify)	98	6.4%	(5.3, 7.8)	6.4	5.5%	(4.6, 6.6)	0.26	
Don’t know/can’t recall	27	1.5%	(1.1, 2.3)	1.5	1.7%	(1.2, 2.4)	0.77	
High blood pressure	341	20.7%	(18.8, 22.9)	20.7	18.6%	(0.9, 16.9)	0.10	
If yes, do you currently take medication for the control of your blood pressure?								
Yes	259	76.6%	(71.6, 80.8)	318	77.3%	(72.8, 81.2)	0.82	
No	82	23.4%	(19.2, 28.4)	94	22.7%	(18.9, 27.2)		
Have you ever received any advice from your doctor or a health professional to reduce your intake of salt/sodium and/or salty foods?								
Yes	350	21.4%	(19.4, 23.6)	454	21.5%	(19.7, 23.3)	0.77	
No	1145	72.9%	(70.6, 75.1)	1550	72.2%	(70.3, 74.2)		
Can’t recall	89	5.7%	(4.6, 7.0)	137	6.2%	(5.3, 7.4)		
Are you the main person who does the grocery shopping in your household?								
Yes	1122	70.2%	(67.8, 72.6)	1493	69.9%	(67.8, 71.9)	0.32	
No	143	9.9%	(8.4, 11.7)	173	8.7%	(7.5, 10.0)		
No, I share the responsibility	319	19.9%	(17.9, 22.0)	475	21.4%	(19.7, 23.3)		

^a^ Demographic characteristics at T1 & T2 weighted to represent Victorian population (Census 2016) for age and sex [32]. ^b^
*p*-value determined via Pearson’s chi-squared test based on weighted data. ^c^ Except where otherwise indicated data taken from the 2016 Victorian Census and reflects the percentage of adults aged 18–65 years residing in Victoria [32]. ^d^ Data taken from the 2016 Survey of Education and Work and includes information on educational attainment in Victorian adults aged 15–74 years. Consistent with our definition of socioeconomic background (SES), we grouped the following responses into each group. Low SES: ‘Year 12 or equivalent’, ‘Year 11′, ‘Year 10′ or ‘Below Year 10′; mid SES: ‘Certificate III/IV’ or Advanced Diploma/Diploma’; high SES: ‘Bachelor Degree’, ‘Graduate Diploma/Graduate Certificate’ or ‘Postgraduate Degree’ [33]. ^e^ T1 *n* = 1568 and T2 *n* = 2128, as participants who responded “don’t know” or “prefer not to answer” for their highest level of education were excluded. ^f^ T1 *n* = 1363 and T2 *n* = 1852, as participants who responded with missing data or “don’t know” or “prefer not to answer” for either height or weight were excluded. ^g^ Data taken from the 2016 Victorian Population Health Survey 2016, estimates based on self-reported height and weight [34].

**Table 2 nutrients-12-01216-t002:** Change in knowledge, attitudes and behaviors related to dietary salt among Victorian adults, 2015 (*n* = 1584) and 2018 (*n* = 2141) ^a^.

Item	2015 (T1)	2018 (T2)	% Difference	*p*-Value ^b^	Adjusted Odds Ratio (95% CI) ^c^	*p*-Value
	% (95% CI)	% (95% CI)				
**Knowledge**						
Knows relationship between salt and sodium	30.6% (28.3, 33.0)	33.5% (31.5, 35.6)	2.9%	0.07	1.12 (0.97, 1.31)	0.12
Knows Australians eat too much salt	83.6% (81.6, 85.4)	82.0% (80.1, 83.6)	−1.6%	0.22	0.92 (0.77, 1.11)	0.39
Knows main source of salt in the Australian diet is from salt from processed foods such as breads, sausages and cheese	72.1% (69.7, 74.3)	75.7% (73.7, 77.6)	**3.7%**	**0.02**	1.22 (1.04, 1.42)	**0.01**
Knows recommended salt intake e.g., 5 g/day	26.6% (24.4, 29.0)	28.8% (26.8, 30.8)	2.1%	0.17	1.11 (0.95, 1.29)	0.17
Knows excess salt could damage health	89.2% (87.5, 90.6)	89.3% (87.8, 90.6)	0.1%	0.92	1.00 (0.81, 1.25)	0.95
Knows adverse effect of eating excess salt on:						
High blood pressure	78.9% (76.7, 80.9)	81.3% (79.5, 83.0)	2.4%	0.08	1.17 (1.00, 1.39)	0.06
Kidney disease	56.8% (54.2, 59.3)	58.7% (56.6, 60.9)	2.0%	0.24	1.09 (0.95, 1.25)	0.23
Heart disease	73.9% (71.5, 76.0)	76.1% (74.2, 78.0)	2.3%	0.13	1.14 (0.98, 1.34)	0.10
Stroke	60.2% (57.7, 62.7)	61.4% (59.2, 63.5)	1.2%	0.50	1.06 (0.92, 1.22)	0.42
Stomach cancer	29.9% (27.6, 32.3)	29.9% (27.9, 32.0)	0%	0.99	1.00 (0.86, 1.16)	0.97
**Attitudes**						
Believes						
Eats less salt than recommended	18.6% (16.7, 20.7)	17.2% (15.6, 19.0)	**−1.4%**	**0.001** ^d^	Not completed	
Eats about the right amount of salt	37.5% (35.1, 40.0)	36.7% (34.6, 38.8)	**−0.9%**			
Eats more salt than recommended	26.0% (23.8, 28.3)	31.5% (29.5, 33.6)	**5.5%**			
*Doesn’t know*	17.9% (16.0, 19.9)	14.6% (13.1, 16.2)	**−3.3%**			
Believes Himalayan salt/sea salt/gourmet salts are healthier than regular table salt	37.5% (35.1, 40.0)	38.2% (36.0, 40.3)	**0.7%**	**<0.001**	1.02 (0.89, 1.18)	0.74
Believes salt needs to be added to food to make it tasty	39.3% (36.9, 41.8)	39.0% (36.9, 41.2)	−0.3%	0.85	0.97 (0.85, 1.12)	0.71
Believes their own health would improve if they reduced salt in their diet	38.6% (36.1, 41.1)	42.8% (40.7, 45.0)	**4.2%**	**0.01**	**1.19 (1.04, 1.37)**	**0.02**
Believes it is hard to understand sodium information on food labels	46.1% (43.5, 48.6)	40.4% (38.2, 42.6)	**−5.7%**	**<0.001**	**0.80 (0.70, 0.92)**	**0.002**
Believes lower salt options in restaurants/cafes/pubs are limited	54.1% (51.5, 56.6)	56.2% (54.0, 58.4)	2.1%	0.21	1.10 (0.96, 1.26)	0.17
Believes there should be laws to limit salt added to manufactured foods	57.6% (55.0, 60.1)	56.8% (54.6, 59.0)	−0.7%	0.67	0.98 (0.85, 1.12)	0.77
**Behaviors**						
Adds salt to food at the table	49.6% (47.1, 52.2)	51.6% (49.4, 53.8)	1.9%	0.26	1.09 (0.95, 1.24)	0.24
Adds salt during cooking	64.3% (61.8, 66.7)	68.1% (66.0, 70.1)	**3.8%**	**0.02**	1.14 (0.99, 1.33)	0.06
Places a saltshaker on the table during meal	49.2% (46.6, 51.7)	46.7% (44.5, 48.9)	−2.4%	0.16	0.92 (0.80, 1.06)	0.24
Trying to cut down on the amount of salt eaten	38.7% (36.2, 41.2)	40.1% (37.9, 42.3)	1.4%	0.41	1.09 (0.95, 1.26)	0.23
Uses food labels to check salt/sodium content ^e^	57.1% (54.6, 59.7)	54.5% (52.3, 56.7)	−2.6%	0.13	0.90 (0.79, 1.04)	0.15
Avoids eating packaged, ready-to-eat foods ^f^	75.3% (73.1, 77.5)	71.9% (69.9, 73.9)	**−3.4%**	**0.03**	**0.84 (0.72, 0.98)**	**0.03**
Uses spices/herbs instead of salt when cooking ^g^	80.3% (78.1, 82.2)	78.4% (76.5, 80.2)	−1.9%	0.19	0.91 (0.77, 1.08)	0.26
Avoids eating from fast food restaurants ^h^	78.1% (75.9, 80.1)	74.3% (72.3, 76.2)	**−3.8%**	**0.01**	**0.81 (0.69, 0.95)**	**0.01**
Avoids eating from Asian style restaurant or takeaway store^i^	67.3% (64.8, 69.7)	63.2% (61.0, 65.4)	**−4.1%**	**0.01**	**0.84 (0.73, 0.97)**	**0.02**
Buys no salt or reduced salt foods^j^	71.6% (69.2, 73.8)	67.3% (65.1, 69.3)	**−4.3%**	**0.01**	**0.83 (0.71, 0.96)**	**0.01**
Asks to have restaurant meals prepared without salt^k^	30.2% (27.8, 32.6)	25.5% (23.6, 27.6)	**−4.6%**	**<0.001**	**0.80 (0.68, 0.93)**	**0.01**

**^a^** Analysis weighted to represent Victorian population (Census 2016) for age and sex [32]. ^b^ Association assessed by Pearson’s Chi-squared test. ^c^ Reference category T1; adjusted for country of birth, diagnosed with chronic condition. ^d^ When response ‘I don’t know’ is removed from analysis the overall finding remains unchanged e.g., ‘I eat less salt than recommended’ −2.5% change, ‘I eat about the right amount of salt’ −2.8% change, ‘I eat more salt than recommended’ +5.3% change, *p* = 0.012. ^e^ T1 *n* =1544 and T2 *n* = 2093 as does not apply to me (T1 *n* = 40 (2.5%) and T2 *n* = 48 (2.2%) excluded from analysis. ^f^ T1 *n* = 1540 and T2 *n* = 2083 as does not apply to me (T1 *n* = 44 (2.8%) and T2 *n* = 58 (2.7%) excluded from analysis. ^g^ T1 *n* = 1523 and T2 *n* = 2066 as does not apply to me (T1 *n* = 61 (3.9%) and T2 *n* = 75 (3.5%) excluded from analysis. ^h^ T1 *n* = 1536 and T2 *n* = 2064 as does not apply to me (T1 *n* = 48 (3.0%) and T2 *n* = 77 (3.6%) excluded from analysis. ^i^ T1 *n* = 1512 and T2 *n* = 2028 as does not apply to me (T1 *n* = 72 (4.5%) and T2 *n* = 113 (5.3%) excluded from analysis. ^j^ T1 *n* = 1542 and T2 *n* = 2092 as does not apply to me (T1 *n* = 42 (2.7%) and T2 *n* = 49 (2.3%) excluded from analysis. ^k^ T1 *n* = 1531 and T2 *n* = 2062 as does not apply to me (T1 *n* = 53 (3.3%) and T2 *n* = 79 (3.7%) excluded from analysis. Bold data indicate difference was statistically significant from T1 vs. T2 *p* < 0.05

**Table 3 nutrients-12-01216-t003:** Change in knowledge, attitudes and behaviors related to dietary salt among sub-sample of parents/caregivers, 2015 (*n* = 544) and 2018 (*n* = 799) ^a^.

Item	2015 (T1)	2018 (T2)	% Difference	*p*-Value ^b^	Adjusted Odds Ratio (95% CI) ^c^	*p*-Value
	% (95% CI)	% (95% CI)				
**Knowledge**						
Knows children eat too much salt	67.8% (63.7, 71.7)	71.4% (68.0, 74.6)	3.6%	0.17	1.20 (0.93, 1.54)	0.16
Knows salt is harmful for children’s health	74.0% (70.1, 77.6)	75.0% (71.7, 78.1)	1.0%	0.68	0.99 (0.76, 1.29)	0.94
**Attitudes**						
Believes more action needs to be taken to reduce the salt in children’s foods	76.6% (72.8, 80.1)	77.7% (74.5, 80.6)	1.1%	0.66	1.01 (0.77, 1.34)	0.94
Believes limiting salt in own children’s food is important	64.3% (60.1, 68.3)	72.6% (69.3, 75.7)	**8.3%**	**0.001**	**1.41 (1.10, 1.81)**	**0.006**
**Behaviors**						
Adds salt to foods prepared for your child/ren	50.2% (45.9, 54.5)	44.3% (40.7, 47.9)	**−5.9%**	**0.04**	**0.76 (0.60, 0.96)**	**0.02**
Places a saltshaker on table at mealtimes	48.1% (43.8, 52.4)	38.3% (34.8, 41.9)	**−9.8%**	**<0.001**	**0.70 (0.55, 0.88)**	**0.003**
Child/ren adds salt to food at the table	35.7% (31.7, 40.0)	26.5% (23.4, 29.9)	**−9.2%**	**<0.001**	**0.67 (0.53, 0.87)**	**0.003**

**^a^** Analysis weighted to represent Victorian population (Census 2016) for age and sex [32]. ^b^ Association assessed by Pearson’s Chi-squared test. ^c^ Reference category T1; adjusted for SES, COB, diagnosed with chronic condition, responsibility for household grocery shopping. Bolded data indicate difference was statistically significant from T1 vs. T2 *p* < 0.05.

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
