# Peer review of "Salt-Related Knowledge, Attitudes and Behaviors (KABs) among Victorian Adults Following 22-Months of a Consumer Awareness Campaign"

_nutrients, 2020, doi:10.3390/nu12051216_

Round 1

Reviewer 1 Report

Thank you for the opportunity to review.

The abstract should include why the problem of salt intake is important for Australian society.

The course of the study is described in detail. I'm interested in:
- why the survey covered 1,584 people in 2015 and 2141 in 2018.
Are these values representative of the studied region?

Discussions should be improved. There are no references to research results from other scientists. The authors only refer to research in Great Britain conducted after a social company in 2004. This is over 15 years ago. For me, the current discussion is about nothing.

The summary should include practical implications. How conducted research helped to solve the problem. What are the author's suggestions for future advertising companies promoting a reduction in the amount of salt in the diet?

Author Response

Reviewer No 1

Comment 1:

The abstract should include why the problem of salt intake is important for Australian society.

Response 1:

An additional sentence to address this has been added to the abstract.

Comment 2:

The course of the study is described in detail. I'm interested in: - why the survey covered 1,584 people in 2015 and 2141 in 2018. Are these values representative of the studied region?

Response 2:

The mid-point (T2) evaluation of the consumer awareness campaign was designed to be assessed from sample populations derived from an online research panel only, through Lightspeed Research. In contrast the baseline survey included additional participants recruited from the community and Facebook (NB these groups will be included in the final evaluation). Therefore number of participants recruited through Lightspeed Research differs between baseline and T2 evaluation (1584 in 2015 and 2141 in 2018) as it is not possible to predict the precise numbers of online responders through Lightspeed and we wished to ensure that we obtained an adequate sample size.

Comment 3:

Discussions should be improved. There are no references to research results from other scientists. The authors only refer to research in Great Britain conducted after a social company in 2004. This is over 15 years ago. For me, the current discussion is about nothing.

Response 3:

As suggested, the discussion has been amended to refer to other relevant research, for example that conducted in South Africa and the UK (lines 366-393, lines 416-417).  We have also compared the results to a recent systematic review documenting salt-related KABs in 24 studies conducted across12 high income countries, which are comparable to the Australian setting (2018 Bhana et al.). It is worth noting that  there have been relatively few large-scale salt reduction consumer awareness campaigns conducted in countries similar to the Australian setting, hence making comparisons of our findings to existing literature is somewhat limited. The evaluation of the Food Standards Agency salt reduction initiative, which was conducted during 2004 to 2009,  is a particularly relevant and major piece of work to make comparisons with. The situation in the UK is similar to that in Australia, with regards to the key sources of salt in the diet and key messages included within each countries consumer awareness campaign. A range of peer-reviewed literature related to evaluation of the UK’s consumer awareness campaign is referenced throughout this revised section.

Comment 4:

The summary should include practical implications. How conducted research helped to solve the problem. What are the author's suggestions for future advertising companies promoting a reduction in the amount of salt in the diet?

Response 4:

This is a useful suggestion. We have included information on this at line 444-449. We have also added a sentence to indicate that the final additional evaluation findings can be used to further inform future salt reduction initiatives (line 450-451).   

Reviewer 2 Report

The study by C. Grimes et al documents the retained or changed salt-related knowledge of Victorian adults 22 months after previous salt related health campaigns and surveys. The general conclusion of most readers would probably be that it is difficult if not impossible to change population health by providing sound scientific evidence that harm can result from unhealthy behavior.

Major concerns: None

Minor Concerns:

  1. The authors should include a caveat that the fact that the survey respondents received a monetary reward for participating may have biased the results. 
  2. The fact that previous publications have demonstrated that up to 11% of individuals of Northern European decent have to increase their salt consumption in order to lower their blood pressure should be stated in the discussion (PMID: 23197156). This condition is referred to as inverse salt sensitivity. This is important since participants self-reported on having a diagnosis of a chronic condition such as hypertension and they reported their use of antihypertensive medication and had received medical advice from a health professional to limit salt intake.  This question is somewhat dated since practicing physicians don’t realize that they are inappropriately medicating their inverse salt sensitive patients when they should be advising them to consume increased amounts of salt to lower their blood pressure into the reference range. Furthermore, the question posed to the volunteers, “‘my health would improve if I reduced the amount of salt in my diet,” should also be noted as dated and not in line with current medical knowledge.

Author Response

Reviewer no 2

The study by C. Grimes et al documents the retained or changed salt-related knowledge of Victorian adults 22 months after previous salt related health campaigns and surveys. The general conclusion of most readers would probably be that it is difficult if not impossible to change population health by providing sound scientific evidence that harm can result from unhealthy behavior.

Major concerns: None

Minor Concerns:

Comment 1:

The authors should include a caveat that the fact that the survey respondents received a monetary reward for participating may have biased the results. 

Response 1:

As suggested this has been acknowledged in the limitations section of the discussion.

 Comment 2:

The fact that previous publications have demonstrated that up to 11% of individuals of Northern European decent have to increase their salt consumption in order to lower their blood pressure should be stated in the discussion (PMID: 23197156). This condition is referred to as inverse salt sensitivity. This is important since participants self-reported on having a diagnosis of a chronic condition such as hypertension and they reported their use of antihypertensive medication and had received medical advice from a health professional to limit salt intake.  This question is somewhat dated since practicing physicians don’t realize that they are inappropriately medicating their inverse salt sensitive patients when they should be advising them to consume increased amounts of salt to lower their blood pressure into the reference range. Furthermore, the question posed to the volunteers, “‘my health would improve if I reduced the amount of salt in my diet,” should also be noted as dated and not in line with current medical knowledge

 Response 2:

We acknowledge that salt sensitivity has been raised as an issue by some previous studies and that it is impractical to test and identify those with salt sensitivity in the population. However, evidence from RCTs clearly demonstrate that reductions in salt intake lead to reductions in blood pressure among those with high blood pressure as well as those in the range of normal blood pressure (Kotchen et al. NEJM 2013;368:13). This evidence forms the basis for population wide strategies to reduce salt intake that seek to shift the distribution of population blood pressure downwards to prevent adverse cardiovascular outcomes. Around the world among leading health agencies there is a consensus to lower the amount of salt consumed in the diet (https://ish-world.com/news/a/WHL-ISH-Salt-Policy-Statement/) as current intakes around the globe are well in excess of dietary recommendations (https://bmjopen.bmj.com/content/3/12/e003733).

Round 2

Reviewer 1 Report

In my opinion, the improvements made are sufficient.  I keep my fingers crossed and wait for the printed version.